# Novel Dry Spinning Process of Natural Macromolecules for Sustainable Fiber Material -1- Proof of the Concept Using Silk Fibroin

**DOI:** 10.3390/ma15124195

**Published:** 2022-06-13

**Authors:** Ryo Satoh, Takashi Morinaga, Takaya Sato

**Affiliations:** Department of Creative Engineering, National Institute of Technology (KOSEN), Tsuruoka College, 104 Sawada, Inooka, Tsuruoka 997-8511, Yamagata, Japan; r-satoh@tsuruoka-nct.ac.jp (R.S.); takayasa@tsuruoka-nct.ac.jp (T.S.)

**Keywords:** dry spinning, silk fibroin, ionic liquids

## Abstract

Researchers around the world are developing technologies to minimize carbon dioxide emissions or carbon neutrality in various fields. In this study, the dry spinning of regenerated silk fibroin (RSF) was achieved as a proof of concept for a process using ionic liquids as dissolution aids and plasticizers in developing natural polymeric materials. A dry spinning equipment system combining a stainless-steel syringe and a brushless motor was built to generate fiber compacts from a dope of silk fibroin obtained by degumming silkworm silk cocoons and ionic liquid 1-hexyl-3-methyl-imidazolium chloride ([HMIM][Cl]) according to a general method. The maximum stress and maximum elongation of the RSF fibers were 159.9 MPa and 31.5%, respectively. RSF fibers containing ionic liquids have a homogeneous internal structure according to morphological investigations. Elemental analysis of fiber cross sections revealed the homogeneous distribution of nonvolatile ionic liquid [HMIM][Cl] in RSF fibers. Furthermore, the removal of ionic liquids from RSF fibers through impregnation washing with organic solvents was verified to enhance industrial applications. Tensile testing showed that the fiber strength could be maintained even after removing the ionic liquid. Thermogravimetric analysis results show that the organic solvent 1,1,1,3,3,3-hexafluoro-2-propanol is chemically coordinated to silk fibroin and, as a natural polymer, can withstand heat up to 250 °C.

## 1. Introduction

As global warming progresses, various weather abnormalities such as droughts, massive typhoons, and localized heavy snowfalls become more frequent [1]. There are concerns that the greenhouse effect will cause permafrost to thaw, releasing unknown frozen viruses (*Morbillivirus*) and causing new infections [2]. Researchers worldwide are developing technologies to reduce carbon dioxide emissions or carbon neutrality in various fields [3,4,5].

In this context, as researchers and engineers of textile materials, we are focusing our efforts on developing new energy-effective spinning processes using renewable natural polymer materials, such as textile and polymer materials, which do not depend on petroleum resources [6]. Since the days when naturally occurring polymeric materials such as cellulose, silk, and wool were used directly as fibers, previous textile chemists developed a wet spinning process of dissolving natural polymers in a solvent and spinning the solution. This is referred to as wet spinning [7]. Wet spinning spider silk protein produced by fungi into fibers is now being researched and developed to produce high-strength, high-toughness renewable fibers by Sekiyama et al. [8]. However, these natural polymers are difficult to dissolve in solvents because of their strong intramolecular and intermolecular hydrogen bonds, which often require complicated dissolution processes such as the use of special solvents containing metal ions or minor chemical modifications to break the hydrogen bonds [9,10]. Therefore, in many cases, designing an ecological and cost-effective manufacturing process is challenging. The difficulty in dissolving the polymer is greater for high molecular weight polymers with good fiber properties [11,12]. However, it has been reported that ionic liquids (ILs) can dissolve natural polymers such as cellulose [13] and silk fibroin (SF) [14], and research is ongoing for using ILs as spinning solvents [15,16,17]. However, due to the high cost of ILs and their nonvolatile nature, the focus of development has been on wet spinning, and little research has been conducted on dry spinning, which is easier, less expensive, and significantly more efficient than wet spinning [18,19]. We have developed a process to obtain natural polymer fibers by dry spinning from a solution of natural polymers dissolved in a solvent containing a small amount of IL as a hydrogen bond cleaver [20]. Typical hydrogen bond cleavers are metal salt compounds dissolved in a solvent that interacts with donor types of substituents such as hydroxyl, carboxylic, and amino groups in the solution to cleave hydrogen bonds [21,22]. During the fiber formation, salt substances crystallize and create voids in the fiber [23]. Using an IL instead of a metal salt compound may prevent the formation of voids [24]. The IL was expected to remain uniformly in the fiber and function as a plasticizer [25]. The nonvolatility of ILs allows them to function as permanent plasticizers and softeners, maintaining the fibers’ flexibility even under vacuum conditions. In this study, we describe the development of a process that uses ILs as a dissolution aid for natural-derived macromolecules, including a proof-of-concept study of dry spinning using regenerated SF (RSF).

## 2. Materials and Methods

### 2.1. Materials

Ethanol, methanol, and calcium chloride (CaCl_2_) were purchased from Kanto Chemical (Tokyo, Japan). Sodium carbonate (Na_2_CO_3_) and lithium chloride were purchased from FUJIFILM Wako Pure Chemical (Osaka, Japan). 1-Hexyl-3-methyl-imidazolium chloride ([HMIM][Cl], ≥95% purity) was purchased from Sigma-Aldrich (St. Louis, MO, USA) as the ionic liquid. 1,1,1,3,3,3-Hexafluoro-2-propanol (HFIP) was purchased from Fluorochem (Derbyshire, UK). All chemical reagents were used without further purification.

Silkworm silk cocoons were kindly donated by Tsuruoka Silk (Yamagata, Japan). Marseilles soap (additive free) was purchased from Miyoshi Soap (Tokyo, Japan). A cellulose dialysis tubing (diameter, 21.4 mm; molecular weight cutoff (MWCO), 12,000–14,000 Da) was purchased from SERVA Electrophoresis (Heidelberg, Germany). Nitrogen (N_2_) gas was purchased from Taiyo Nippon Sanso (Tokyo, Japan). Water purification systems (PURELAB flex 3, ELGA LabWater, High Wycombe, UK) were used to obtain ultrapure water.

### 2.2. Equipment

An SSY-30E stainless steel cylinder (23 mm i.d.) was purchased from Musashi Engineering (Tokyo, Japan). A UNP-23 dispenser nozzle (0.3 mm i.d.) was purchased from Unicontrols (Chiba, Japan). Brushless direct current electric motors (model GFV2G20) and the speed control units (model BMUD30-A2) were purchased from Oriental Motor (Tokyo, Japan). Stainless-steel filter membrane (vertical mesh/horizontal mesh, 500/3600; filtration threshold, 5 µm) was purchased from Yao Kanaami (Osaka, Japan).

### 2.3. Silk Degumming

Degumming was executed with minor modifications according to the literature [26]. Using a temperature-controlled water bath, 5.0 g of silkworm silk cocoons and 500 mL of alkaline solution (0.25 *w*/*v*% Marseilles soap, 0.25 *w*/*v*% Na_2_CO_3_) were boiled at 85 °C for 15 min. After removing the solution by decantation, the residue was rinsed six times with ultrapure water at 80 °C and dried in vacuo overnight. In 100 g of Ajisawa’s solution [27] (CaCl_2_/water/ethanol = 1:8:2, molar ratio), 15 g of degummed silk was dissolved. The degummed silk was thoroughly blended with Ajisawa’s solution and heated at 55 °C for 1 h. The silk solution was filtered through a stainless-steel filter membrane. The filtrate was dialyzed against ultrapure water immediately using cellulose dialysis tubing (MWCO: 12,000–14,000 g/mol) at 4 °C for 4 days until the conductivity was lower than 10 mS. The dialyzed solution (diluted with ultrapure water up to 4 *w*/*v*%) was then lyophilized, and the resulting SF was desiccated and stored at ambient temperature.

### 2.4. Preparation of Dope Solution

To 3.0 g of SF, 16.25 g of HFIP was added, and the mixture was heated at 50 °C for 3 h. Subsequently, 0.75 g of [HMIM][Cl] was added dropwise, and the mixture was heated at 50 °C for 1 h. The final dope composition was SF/[HMIM][Cl]/HFIP = 15:3.75:81.25 (*w*/*w*/*w*). The resulting IL-containing SF dope solution was brought to ambient temperature and used for dry spinning.

### 2.5. Dry Spinning and Heat Stretching

A schematic of the dry spinning equipment is shown in Figure 1a. Under N_2_ pressure, the dope solution in a stainless-steel cylinder (23 mm i.d.) with a dispenser nozzle (0.3 mm i.d.) was discharged into the air. The distance between the spinneret and the first guide roller was 100 cm. The pressure was controlled between 0.3 and 0.02 MPa to stabilize the spinning operation. Simultaneously, the roller winding speed was reduced from 12.0 to 4.0 m/min. The thread was predried among the guide rollers shortly after discharging and led to the take-up roller (4.2 m/min) as shown in Figure 1a. The resulting solidified thread was dealt as the as-spun fiber. The obtained yarns were dried in vacuo at ambient temperature for >5 h with bobbins.

The equipment for heat stretching is shown in Figure 1b. The speeds of the let-off and take-up roller were set to 0.25 and 0.88 m/min, respectively (3.5-fold stretch). A hot plate was located between the two rollers and maintained at 120 °C. The obtained heat-stretched fiber was dried in vacuo for >5 h. To examine the physical properties of the fiber with or without the IL-washing process, the heat-stretched fiber was soaked in methanol to remove [HMIM][Cl].

### 2.6. Characterization

Scanning electron microscopy (SEM) was performed using a JSM-7100F field-emission scanning electron microscope (JEOL, Tokyo, Japan). The acceleration voltage was 2 kV. Elemental analysis of the fibers was performed using a JED-2300F energy-dispersive X-ray spectroscopy (EDX) instrument (JEOL) equipped with a solid-state detector. An Instron Model 3342 single-column testing system (Instron, Norwood, MA, USA) was equipped with grips with high frictional faces for flats, and a static load cell (maximum load, 500 N) was used for tensile testing. Instron Bluehill Lite Software (version 2.28.832) was used for data collection. Microsoft Excel (version 2202, Microsoft, Redmond, WA, USA) was used to process the data. The fiber specimens (randomly taken) were cut and fixed to paper mounts. The initial gauge length was 20 mm. The elongation rate was 2 mm/min (10% initial length). Using Microsoft Excel, the cross-sectional area of the fiber was determined from the cross-sectional images of the SEM results. A Thermo Plus TG8120 (Rigaku, Tokyo, Japan) was used to perform thermogravimetric (TG)–differential thermal analysis (DTA). Data acquisition was performed at a temperature of 50–400 °C, the rising rate was 10 °C/min, and the flow rate of N_2_ gas was 200 mL/min. The sampling time was set at 1.0 s. Al_2_O_3_ was used as the reference material.

## 3. Results and Discussion

### 3.1. Morphological Observation

SEM was used to clarify the surface and cross-sectional images of as-spun (Figure 2a,b) or heat-stretched (Figure 2c,d) RSF fibers. The RSF fibers before/after heat stretching had homogeneous internal structures, contrary to the possibility of forming a heterogeneous sea–island structure. A belt-like flat fiber cross section was observed for both the as-spun and heat-stretched fibers.

From the morphology inside the cross-section of the fibers, the void formation could be effectively prevented using ILs instead of metal salts. Because no rapid volatilization of HFIP occurred during the first solidification process, all the RSF fibers had smooth surfaces. The authors considered that belt-shaped RSF fibers were formed and caused by tensile tension from the guide rollers during solidification into fiber bodies or shrinkage tension from drying between the guide rollers. To investigate the more intrinsic mechanical properties, a circular fiber needed to be used as a test specimen for tensile testing. Therefore, extending the vertical distance from the spinneret to the first guide roller and extending the time for HFIP volatilization was effective. The plasticity of nonvolatile ILs is high during the drying process despite the belt-shaped formation. Furthermore, the thermo-melting-like state of the RSF fibers in heat stretching allowed us to achieve further deformed and modified fibers that had star-shaped, Y-shaped, or other odd-shaped cross sections. In the case of the RSF fibers in this research, it may have been possible to develop from dry spinning to melt spinning.

### 3.2. Elemental Analysis

Figure 3 shows the distribution of ILs in RSF fibers. Elemental mapping revealed the distribution of chlorine atoms specifically in the chemical structure of [HMIM][Cl] using EDX analysis. The uniform distribution of chlorine atoms in the cross-section corresponding to the SEM images was confirmed in the picture.

### 3.3. Removal of IL from the RSF Fibers

Figure 4 shows the EDX spectra of heat-stretched RSF fibers before and after methanol immersion. In addition, the extracted EDX spectrum of the RSF fiber after heat stretching is shown in Figure 4a, as well as the 2.6 keV indicated chlorine atoms from [HMIM][Cl]. Figure 4b shows the extracted EDX spectrum of the RSF fiber that was washed with methanol after heat stretching, and chlorine was not confirmed.

After heat stretching, IL was easily removed using a washing process that involved immersing the fibers in a suitable organic solvent. This process suggested the possibility of not only removing ILs used specifically as forming aids from RSF fibers as needed but also controlling the amount of IL expected to function as a plasticizer. Furthermore, under solution equilibrium, the immersion process was expected to allow for the exchange and distribution of the ILs eventually remaining in the RSF fibers.

### 3.4. Mechanical Strength of the RSF Fibers

Figure 5 shows the stress–strain (S–S) plots of heat-stretched RSF fiber (closed circles, ●) and the plots washed with methanol (open squares, □) to compare the mechanical strength of the RSF fiber from a practical viewpoint. Methanol was selected as a typical solvent for ILs. The IL-washing process after heat stretching was designed for industrial application, and methanol was selected as a typical solvent for [HMIM][Cl], which was added as a plasticizer. The main physical values that characterize the RSF fibers in the tensile testing are shown in Table 1. For non-washed (closed circle plots, ●) and washed-off IL (open square plots, □), corresponding to Figure 5, the tensile stress at the point of the maximum load was 159.9 and 105.4 MPa (tensile strain: 3.6% and 2.3%, respectively), the fracture strain was 31.5% and 35.8%, Young’s modulus was 4.86 and 4.83 GPa (to the top yield), and toughness was 43.2 and 39.4 MJ/m^3^ (where calculated from the area under the curve integration with a trapezoidal approximation of the plots), respectively.

The process of washing off the IL did not cause the significant degradation of fiber properties according to the S–S plots after heat stretching. The tensile strain at the yield point was reduced by 1.3% points when methanol was used. This is reasonably explained by the loss of IL, which allowed it to function as a plasticizer. Conversely, the fracture strain increased by 4.3% points in heat-stretched RSF fibers (closed circles (●) in Figure 5) compared with that in the RSF without IL washing (open squares (□) in Figure 5). These results were considered a trade-off between the increased secondary structure of SF protein from an alcohol-induced *β*-coil structure [28] in SF molecules and the loss of flexibility due to the removal of IL as a plasticizer.

The authors hypothesized that fibers containing ILs were momentarily fractured because of an imbalance between the relaxing fiber’s body caused by the plasticizing function of ILs and the fracture of microscopic defects caused by elongation, in which the tearing force prevailed. The methanol-washed fibers lost ILs; however, the secondary structure was increased overall, as described in the literature [28]. The SF molecules themselves exhibit more elongation, as shown in the data. There would be no relaxation in the elongation mechanism as there would be in the presence of the plasticizer, and an instantaneous fracture of the microscopic defects in the fiber was occurring.

### 3.5. Thermophysical Properties

Figure 6 shows the TG–DTA curves of as-spun RSF fiber. Data showed a 2.0 wt% loss relative to the initial weight from 50 °C to 100 °C, with a further 5.1 wt% loss in steps from 110 °C to 185 °C. Desorption of equilibrated moisture from the atmosphere would account for the initial 2.0 wt%. Because the fiber already contains ILs that interact with hydrogen bonds among SF molecular chains, it could be assumed that the moisture was volatilized by heating below 100 °C. Furthermore, the weight loss of approximately 150 °C suggested that the heating desorption of HFIP (boiling point 58 °C for a single chemical entity) and the desorption of HFIP from SF materials at a temperature greater than the boiling point had been considered as the formation of specific hydrogen bonds between SF and HFIP [29,30]. The 10 wt% loss temperature, except for the mass of volatile components (water and HFIP), was 258 °C, and the decomposition temperature was observed in the range of 260–320 °C. The temperatures corresponding to the maximum degradation rate are 285.43 °C (random coil), 281.38 °C (silk-I structure), and 313.73 °C (silk fiber) according to the literature [31]. The current findings backed up these claims.

## 4. Conclusions

In this study, the dry spinning of SF was shown as a proof of concept for a process that uses ILs as dissolution aids and plasticizers in the development of natural polymeric materials. The mechanical properties of the RSF fibers reached 159.9 MPa and 31.5% at maximum stress and elongation, respectively. ILs were uniformly distributed in the fiber, indicating that they can effectively function as a plasticizing additive. Further studies are needed to clarify the function–property relationship between natural polymeric materials and ILs. Currently, there is no way to produce such materials more cheaply than petroleum-derived materials. However, in the future, protein materials may be produced more cheaply than petroleum-derived materials. In such a development situation, the concept of this research, which pioneered a new method of producing raw materials obtained from natural sources, will become even more important.

## 5. Patents

The research reported in this article is based on the following patent: Sato, T.; Morinaga, T.; Satoh, R. Method for producing polymer substance molding. Japan patent JP2021028434A. Publication date, 25 February 2021.

## Figures and Tables

**Figure 1 materials-15-04195-f001:**
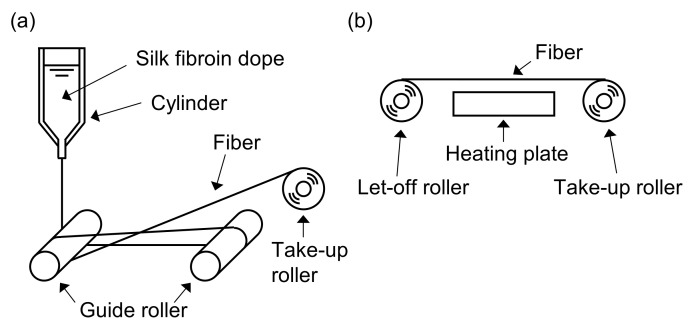
Conceptual images of dry spinning and heat stretching for regenerated silk fibroin. (**a**) Dry spinning system. The extruded dope solution was dried in ambient air to mold into the fiber on the guide rollers equipped with polytetrafluoroethylene tubes to prevent sticking of the dope in process and collected to the take-up roller. (**b**) Heat stretching system. The molded silk fiber was sent from the let-off roller and collected to the take-up roller directly. Spinning fiber was heated for 3 mm tolerance over the heating plate and stretched among the two rollers.

**Figure 2 materials-15-04195-f002:**
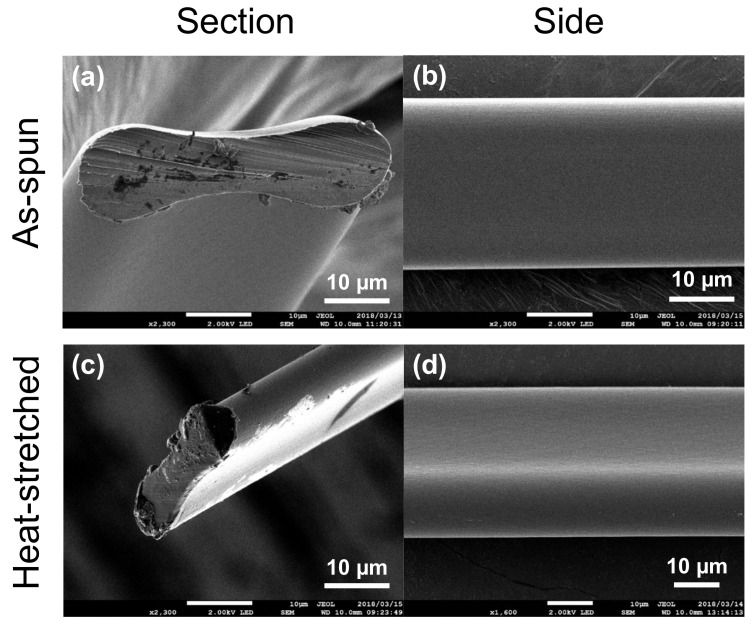
SEM images of regenerated silk fibroin (RSF) fibers: (**a**) sectional and (**b**) side images of RSF as-spun fibers; (**c**) sectional and (**d**) side images of RSF heat-stretched fibers. Each picture has a 10 µm scale bar. The cross-sections were made by direct immersion to liquid nitrogen and cutting with a razor blade.

**Figure 3 materials-15-04195-f003:**
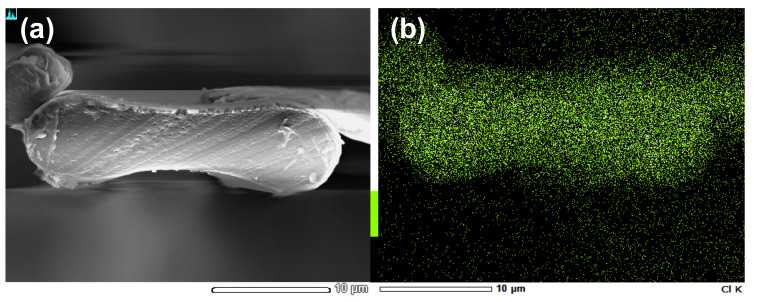
SEM picture (**a**) and an EDX mapping image (**b**) of regenerated silk fibroin fibers. Areas of dense green dots indicate the presence of chlorine atoms. Each picture has a 10 µm scale bar.

**Figure 4 materials-15-04195-f004:**
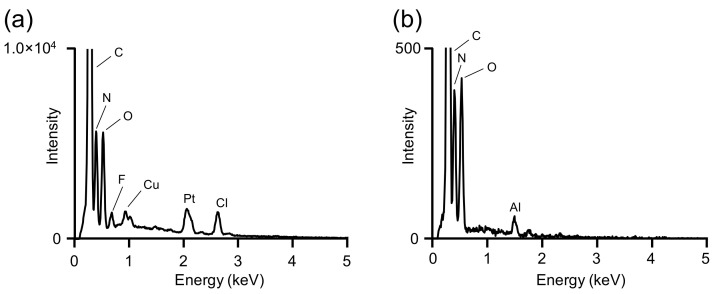
EDX spectra for heat-stretched regenerated silk fibroin (RSF) fibers with/without washing using methanol to confirm the remaining ionic liquid. The RSF fiber (**a**) without washing and (**b**) with washing using methanol. The peaks of copper, platinum, and aluminum in the picture are backgrounds from sample holders or sputter deposition in sample preprocessing.

**Figure 5 materials-15-04195-f005:**
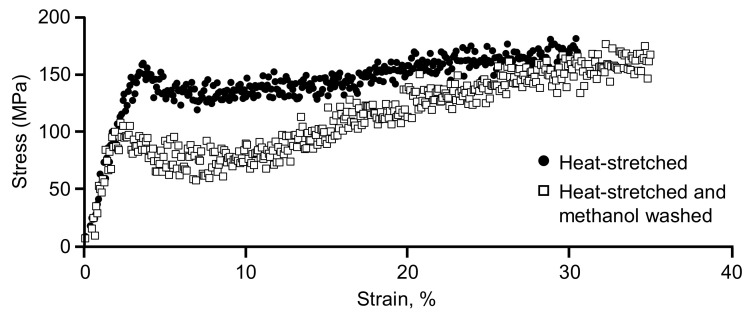
Stress–strain (S–S) plots for heat-stretched regenerated silk fibroin (RSF) fibers. Closed circles (●) represent S–S plots of heat-stretched RSF without wash using methanol. Open squares (□) represent S–S plots of heat-stretched RSF after wash using methanol.

**Figure 6 materials-15-04195-f006:**
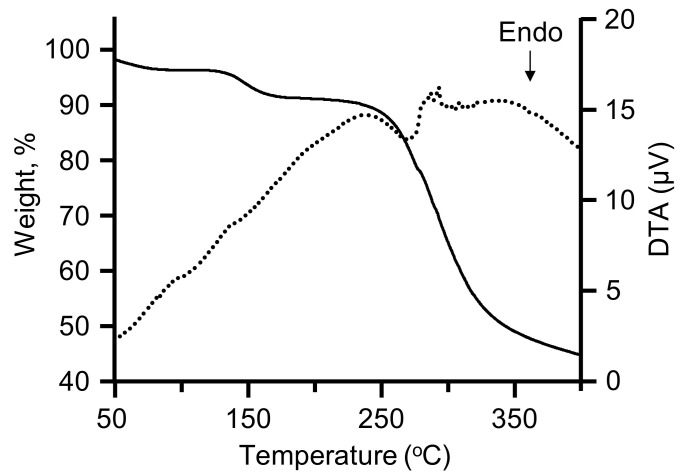
Thermogravimetric–differential thermal analysis curve of as-spun (nondrawing) regenerated silk fibroin fiber.

**Table 1 materials-15-04195-t001:** Mechanical property values of heat-stretched regenerated silk fibroin (RSF) fibers on the tensile testing.

Heat-Stretched RSF	Top Yield (MPa)	Fracture Strain (%)	Young’s Modulus (GPa)	Toughness (MJ/m^3^)
Non-washed (containing IL)	159.9	31.5	4.86	43.2
Washed off IL	105.4	35.8	4.83	39.4

## Data Availability

The data that support the findings of this study are available from the corresponding author, T.M., upon reasonable request.

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
