# Peer review of "Novel Dry Spinning Process of Natural Macromolecules for Sustainable Fiber Material -1- Proof of the Concept Using Silk Fibroin"

_materials, 2022, doi:10.3390/ma15124195_

Round 1

Reviewer 1 Report

you should add some numerical values of results obtained  to the abstract

clarify the novelty in your work and what is the problem you try to solve in this work in the view of the previous research works

Report the physical and chemical properties of the materials used

To decide if the change in values of different mechanical properties is a significant or not you should perform statistical analysis for the results reported in table 1

Reviewer 2 Report

Reviewer comments (materials)

This manuscript describes “Novel dry spinning process of natural macromolecules for sustainable fiber material-1-Proof of the concept using silk fibroin”. This is an interesting and well written article. In this study, experiments are well designed and conclusive. However, there are some minor issues in the current manuscript, and it can be considered for publication after addressing following concerns.

Major and Minor concerns:

  • This manuscript is poorly referenced. Authors need to add more references.
  • Introduction needs more relevant and some recent references.
  • Can authors explain why they explored only one ionic liquid 1-hexyl-3-methyl-imidazolium chloride?
  • Conclusion needs to rewrite with more clarity with supporting experiments and its results.
  • All reference should in uniform pattern.

Reviewer 3 Report

There is a weak introduction part in the manuscript, the topic of obtaining fibers from silk is quite extensively researched but it is poorly disclosed here. With the very brief content of the manuscript and introduction, the whole (first) paragraph is completely irrelevant to the topic of the study.

1) Lines 33-35 The authors' point of view is quite naïve because all fiber types have been needed for various industries. Is it possible to produce cheap silk fibers in volumes equal to equivalent synthetic ones? For example, worldwide PET fiber production has been more than 50 megatons per year.

2) Line 22 It is a very strong statement. Indicate studies in which TGA data show that the solvent is chemically coordinated to the polymer.

3) Line 98 g/mol is preferable because the dimensions must be given in SI units.

4) Figure 1. Which was the distance between the spinneret and the first guide roller?

5) Line 127 An accelerated voltage and a detector type should be given.

6) Line 144 How cross-sections were made? It must be written.

7) Figure 2 Have you experimented with the effect of spinning conditions on fiber shape? How does the shape of the fiber change with air temperature and the distance between the spinneret and the take-up roller? It is important to understand whether this shape is obtained when a gel fiber gets on the roller and flattens it, or it is due to the evaporation process and the formation of a hard shell.

8) Figure 2 What is the cause of such large defects in the fiber in this case?

9) Why didn't you evaluate the degree of polymer orientation by direct X-ray diffraction analysis?

10) Section 3.4. The conclusions of the authors are very controversial. Rather, it appears that the fiber relaxes as the IL is washed out, resulting in a decrease in the degree of orientation, which explains the increase in elongation at break and the decrease in strength values.

11) Figure 6.  Why were the washed fibers not examined? A comparison would be interesting.

In general, the work is a series of experiments with a very weak level of discussion of the results obtained and requires significant improvement in all sections.

Round 2

Reviewer 1 Report

No comment 

Reviewer 3 Report

The authors answered my questions and the text and results obtained are clearer now.

Nevertheless, the discussion and conclusion part especially should be improved.

Line 78. Here is "Da" still used.

All changes in the manuscript must be marked by color.